# Formative Evaluation of Open Goals: A UK Community-Based Multi-Sport Family Programme

**DOI:** 10.3390/children7090119

**Published:** 2020-09-01

**Authors:** Leanne Burton, Kathryn Curran, Lawrence Foweather

**Affiliations:** Physical Activity Exchange, Research Institute for Sport and Exercise Sciences, Liverpool John Moores University, Liverpool L3 5UX, UK; l.r.burton@ljmu.ac.uk (L.B.); k.curran@ljmu.ac.uk (K.C.)

**Keywords:** formative evaluation, physical activity, community, intervention, family, park

## Abstract

Community parks provide opportunities for physical activity (PA) and facilitate social interactions. This formative evaluation assesses the implementation of ‘Open Goals’ (OG), a novel multi-sport programme aiming to increase family PA and community cohesion, delivered weekly by Liverpool Football Club’s charitable foundation to local parks in Liverpool, North West England. Three Open Goals parks were chosen for the evaluation settings. Formative evaluation measures included: System for Observing Play and Recreation in Communities (SOPARC) observations (*n* = 10), direct session observations (*n* = 8), semi-structured interviews with Open Goals coaching staff (*n* = 3), and informal feedback from families (*n* = 5) about their experiences of Open Goals. Descriptive statistics and thematic analysis were applied to quantitative and qualitative data, respectively. Within the three evaluation parks, Open Goals reached 107 participants from May–July 2019, through 423 session attendances. Fidelity of the programme was high (M = 69% of session content delivered as intended). Overall park use when OG was offered compared to when it was not offered was not statistically significant (*p* = 0.051), however, target area use was significantly increased (*p* = 0.001). Overall physical activity levels in parks were significantly (*p* = 0.002) higher when Open Goals was being offered, compared to when it was not. Coaches reported that engagement in OG positively affected family co-participation and children’s behavioural development. Contextual issues included environmental and social barriers to programme engagement, including the co-participation element of the programme and criticism of the marketing of OG. It is evident that community-based multi-sport PA programmes endorsed by professional football clubs are well positioned to connect with local communities in deprived areas and to encourage PA and community engagement. This study suggests that such programmes may have the ability to improve park usage in specific areas, along with improving physical activity levels among families, although further research is required. Effective marketing strategies are needed for promotional purposes. Upskilling of coaches in the encouragement of family co-participation may support regular family engagement in PA in local parks.

## 1. Introduction

The positive impacts of physical activity (PA) on children’s physical and mental health outcomes are well established [1,2,3]. Despite this, many children in England remain insufficiently active, with only 17.5% of children and young people meeting the current PA guidelines of 60 min of moderate-to-vigorous physical activity each day [4]. Beyond this, PA levels decline through childhood and into adolescence, particularly outside of school time and at weekends [4,5,6,7]. Inactivity in childhood tracks into adulthood, increasing associated risks of obesity, cardiovascular disease and diabetes [8]. The development of interventions to promote and maintain children’s physical activity levels is therefore a public health priority.

Family-based physical activity interventions present a much-needed opportunity to increase children’s PA levels, with the family environment and the community in which children live believed to have influences on their associated health behaviours [7,9]. Current evidence suggests children are more active during school-time and it is therefore important to target PA at the weekends and outside of school [10,11,12]. Family-based PA interventions have been developed and evaluated with varying success and little is known about the most effective way to involve families in PA interventions [1,2]. Evidence suggests that focusing an intervention on something other than PA for health or weight loss may be a valuable approach. For example, spending time with family, developing movement skills and improving confidence and self-esteem may be more desirable outcomes for an intervention, using PA merely as a vehicle for such change [1]. Family-based PA interventions are predominantly delivered via school settings [1]. More research into whether family-based interventions outside of school can increase PA is required.

There are over 27,000 green spaces, neighbourhood parks and outdoor recreational facilities across the UK that are appropriate settings for families to participate in physical activity [13]. Access to and use of green spaces and park areas has not only been associated with physical health, but also with improved mental health and social wellbeing in children and adults [14,15]. ‘Doorstop green space’ is used to refer to neighbourhood and local parks accessible to many residents of UK conurbations. Arguably, these spaces have the greatest ability to influence health behaviours, given their relative ease of access and potential incidental use through residents’ daily movements in their local area [16,17,18]. Although there are several barriers to park use in families and park-based PA, poor quality and perceptions of safety are some of the reasons for low usage [19]. Within parks, poor maintenance and evidence of neglect (i.e., vandalism and graffiti) increases the likelihood of further misuse and behaviour considered to be anti-social [16]. Evidence suggests that people in deprived communities value local park space, but cite anti-social behaviour as a deterrent from their use [20,21]. Further, studies conducted overseas indicate that parks are often underutilised [22] and are not used to their maximum potential for participation in PA, both in child and adult populations [23,24,25,26]. Therefore, intervention strategies that increase the accessibility and perceived safety of parks are required, and such interventions may provide opportunities for families to engage in PA in park settings. 

In light of this evidence, a programme entitled Open Goals was piloted in 2017 by the Liverpool Football Club Foundation, the club’s official charity. The ongoing programme was designed to provide free, family-based PA opportunities in local community open spaces and parks, with the desire to increase family time spent together; to improve social, mental and physical wellbeing; and to develop fundamental movement skills, as well as increasing community cohesion and park use.

This study aimed to conduct a formative evaluation of the Liverpool Football Club (LFC) Foundation’s Open Goals programme. Formative evaluation is “a rigorous assessment process designed to identify potential and actual influences on the progress and effectiveness of implementation efforts” [27] (p. S1). The most basic purposes of formative evaluation are to maximise the effectiveness of implementation processes, and to provide contextual explanations for the success (or otherwise) of interventions [27]. The evaluation of multi-sport programmes is increasingly used to explore the implementation, feasibility and acceptability of such interventions [28,29,30,31,32]. Findings from this study will be used to aid understandings of the feasibility and challenges of delivering multi-sport PA interventions, as well as to inform interpretations of intervention effectiveness and development of the programme for long-term sustainability [29].

## 2. Materials and Methods

### 2.1. Study Design

Physical Activity Exchange researchers at Liverpool John Moores University were commissioned by Liverpool FC Foundation to undertake an independent evaluation of Open Goals. The evaluation was informed by the Medical Research Council (MRC)’s framework for evaluation of complex interventions [29]—Bauman and Nutbeams’s (2013) [33] evaluation guide and Linnan and Steckler’s (2002) [34]. The evaluation focused on delivery of the intervention to gain an understanding of what activities were being delivered, how they were being delivered, why and to whom they were being delivered. The evaluation also explored contextual elements of the programme through interviews with coaches to help provide an understanding as to how and why the programme has achieved, or failed to achieve, its objectives. The study was conducted from May–July 2019. Ethical approval for this study was obtained by the Liverpool John Moores University Research Ethics Committee (reference: 19/SPS/021).

### 2.2. Participants and Recruitment

A convenience sampling method was used to recruit participants for the research study. In May 2019, all coaches who delivered Open Goals on a regular basis (*n* = 12) were invited to participate in the study via email and gave informed consent prior to taking part. Families (*n* = 5) were recruited from two of the three evaluation parks. Families participating regularly in Open Goals sessions were eligible to participate. Parents/guardians received a letter containing a parent and child information sheet, including contact information for the research team to discuss the project further. Informed consent was sought from all participants prior to data collection. During data collection, families were given the opportunity to withdraw from data collection at any point.

### 2.3. Intervention

The Open Goals multi-sport programme is funded by the Premier League Charitable Fund and has been delivered by the Liverpool FC Foundation since 2017. Open Goals primarily aims to encourage children (aged 5 years and over) and families to be more active, with secondary aims including improving social, mental and physical wellbeing; developing children’s fundamental movement skills; as well as increasing community cohesion and park use. Open Goals (OG) is currently implemented in thirteen parks and open spaces across Liverpool and Wirral, having expanded from 5 parks on inception in 2017 to 9 parks in 2018 and 11 parks in 2019. Parks are chosen in collaboration with ‘Friends of Parks’ (a community organisation) and Liverpool City Council, with input from Merseyside Police, and are situated in areas which these stakeholders view as of high socio-economic deprivation; where crime and anti-social behaviour within the parks is prevalent, and where current physical activity/sports provision is low. This evaluation focused on three of the thirteen current delivery parks, as OG was initiated in these parks at the time of the research study (May 2019).

Coaches were recruited by the LFC Foundation to deliver the programme, with further support provided by local student volunteers. All staff are qualified to deliver a variety of sports programmes and have National Governing Body (NGB) qualifications, with one coach per session being Level 2 Sports Coaching qualified. Staff recruited had previous experience of delivering programmes in community settings. All recruited coaches were required to attend a free educational course, utilising a pre-designed coach education programme from Create Development™.

Marketing materials for OG were distributed through multiple communication channels, including flyers and posters in local community settings (i.e., GP surgeries, local schools) and through social media outlets (Twitter and Instagram). Sample sessions (one per school class) and whole school assemblies, aiming to promote OG and recruit target participants, were also delivered by Open Goals coordinators in primary schools within proximity of an Open Goals park. Sample sessions, commonly referred to in the UK as taster sessions, aim to introduce potential participants to OG coaches and demonstrate the types of activities that are offered during a typical Open Goals session.

OG sessions are structured for delivery in ten-week blocks and run for 60–90 min each week for 52 weeks of the year, taking place on evenings after school or at various times during the weekend, during daylight hours. Park A sessions took place on a weekend and parks B and C sessions took place after school on separate weekday evenings. This formative evaluation study focused on a 10-week block that commenced in three parks in February 2019. Core components to session delivery included; (i) warm up/walk and talk with participants, (ii) skill development, (iii) multi-sport game/activity relating to skill, (iv) involvement of family members and (v) cool down including reflection on session outcomes and ‘teachable moments’ based on Liverpool Football Club’s core values (ambition, commitment, unity and dignity). Some flexibility in the sessions was allowed, supporting coaches to be creative with the games/activities included in the sessions.

#### 2.3.1. Measures and Procedures

Implementation and preliminary effectiveness of the Open Goals programme was assessed using a mixed-methods formative evaluation design. Table 1 summarises the measures and data collection techniques used. Reach was explored through participant registration forms completed by LFC Foundation coaches at sign-up by all OG participants. Demographic information was collected including participant date of birth, ethnicity, sex and postcode. Parents provided this information on behalf of children.

‘Dose delivered’ was assessed through session observations at each park (*n* = 8 sessions) and by coaches recording sessional data (i.e., delivery issues, session length and participant numbers) on a weekly basis. Sessions observed by the researcher also assessed coach delivery using a bespoke observational record form for analysis (see Table 2). The observation form recorded session length; child/parent participant numbers and engagement; how the sessions were introduced, explained and delivered; as well as some questions concerning coach enthusiasm and encouragement. Similar observational techniques have been used in previous studies to evaluate interventions [35,36].

A fidelity score was also calculated for each session observation and converted to a percentage to allow comparison between sessions ((number of components delivered as intended ÷ number of components for session type) × 100). Fidelity was scored as low (≤33%), average (34%–66%) or high (≥67%), as categorised in previous research [37]. For the fidelity of an OG programme to be defined as acceptable, at least two thirds (67%) of the session had to be delivered as intended.

To explore the effectiveness of Open Goals through its influence on park usage by families in the local area, direct observations of park visitors were conducted using a modified version of the System for Observing Play and Recreation in Communities (SOPARC) [38]. This is a reliable, objective observation tool for assessing physical activity in community settings, and previous studies have used SOPARC to assess physical activity in parks [39].

At each park, a trained observer conducted observations of clearly defined target areas. Target areas were pre-determined and included all standard park features likely to provide opportunities for park users to be physically active, including green spaces, playgrounds, pathways and sport specific areas (i.e., football pitch, cricket green). During each scan the observer recorded each individual in view within their target area, according to their broad age group, sex and the activity they were engaged in. In line with original SOPARC protocol [38], physical activity levels were coded as sedentary (i.e., lying down, sitting or standing), walking, or vigorous (i.e., any activity level more strenuous than walking). Data were collected on ten occasions, including five when Open Goals was being offered and five when it was not. For example, in parks where sessions took place during a weeknight, the observation took place from 4–5 pm, the first hour of the session delivery time. On days with adverse weather conditions (i.e., rain, extreme wind, cold, etc.) and on holidays, observations were rescheduled to take place during a subsequent week so that observation data were only collected during optimal conditions. To aid the reliability of the data, a sub-sample (*n* = 4) of observations (both coach and SOPARC observations) were cross-checked. All reliability observations had 100% agreement, indicating strong inter-rater reliability for all recorded characteristics.

To assess the perceived acceptability of the programme, feedback was sought through informal discussions with participating families (*n* = 5). Key areas of conversation included perspectives on the activities delivered, coaching practices and views for improvement of the programme. A question schedule ensured required topics were covered but also allowed families to respond freely. Responses were recorded through notes taken by the first author.

Semi-structured interviews explored coaches’ (*n* = 3) views on the relevance, implementation and perceived impact of the OG programme, to provide some context on the programme from the delivery team. Twelve coaches were invited to take part (25% response rate). Key areas for discussion included perceived impact on family physical activity levels, health and wellbeing; impact on the local community; marketing and session delivery; and barriers and suggestions for improvement of the programme. Interviewees were also given the opportunity to comment on any topics that were not covered by the interview schedule. Interviews were audio recorded and lasted between 25 and 35 min.

#### 2.3.2. Data Analysis

Data from participant registration forms, delivery logs and session observation records were entered into Excel spreadsheets to calculate descriptive statistics. The effects of OG on park usage and physical activity were analysed in SPSS (Version 26, IBM, New York, NY, USA) using independent sample *t*-tests and Chi square tests, respectively.

Participant feedback obtained through informal conversation and collated in researcher field notes was entered into the spreadsheet in order to identify themes. Coach interviews were transcribed verbatim for the purpose of analysis. Thematic analysis was conducted by hand, as described in Braun and Clarke’s six-phase approach [40]. Phase one, familiarisation with the data, involved reading and re-reading of transcripts, while noting down any key points and ideas. Phase two (generating initial codes) involved coding key features of the data systematically and gathering further evidence relevant to each code. Phases three through to five involved the iteration of this process, generating broader themes and ensuring coherence of data. Themes were presented by the first author (L.B.) to the wider team (L.F., K.C.), allowing for alternative interpretations of the data to be discussed and themes to be refined accordingly. Phase six involved the writing up of these themes, with master themes presented as subheadings within the Results section.

## 3. Results

### 3.1. Reach: Are Families Engaging in the Programme?

Open Goals reached 107 different participants in the three evaluation parks between May–July 2019, with 423 session attendances. Individual park breakdowns of participants (30.8% female, 85% White British) are shown in Table 3.

Consideration was afforded by coaches, during the interview process, as to why Open Goals may or may not be reaching the numbers of participants that it expected to. Bad weather throughout the winter and early spring months proved a barrier and affected attendance numbers. Coaches also perceived that the “bad reputation” in some of the parks had an influence on participant numbers due to perceived sense of safety when using the park.

### 3.2. Dose Delivered: How Many Open Goals Sessions Were Delivered?

During the reporting period, 10 sessions were delivered at all three parks, out of 10 planned sessions. When observation of Open Goals sessions took place on eight occasions (Park A = 2, Park B = 3, Park C = 3), 72 children attended, with 43 adult participants. The observed sessions lasted on average 60 min, ranging from 55–70 min.

Open Goals sample sessions were delivered, as part of the Open Goals marketing campaign, in one primary school near park B. One whole school assembly was delivered to all pupils (*n* = 780) in addition to six days of sample sessions across a three-week period, targeting 130 pupils per day (*n* = 780). Parks A and C did not receive the assembly or sample session marketing campaign due to difficulties in accessing the local schools in the area.

### 3.3. Fidelity: To What Extent Is Open Goals Being Delivered as Intended?

Table A1 shows data for the fidelity of Open Goals delivery recorded during observations of coaching sessions (*n* = 8). Average overall fidelity score across parks was 69%, with park C scoring highest for fidelity (M = 77%). In all parks, the welcome aspects of the session scored highest for fidelity, including staff arriving on time (100%), the greeting of participants (100%) and completion of registration forms and risk assessments (100%). Three key aspects of sessions scored particularly low on fidelity in several sessions, including involvement of adult family participants (50%), encouragement of families to seek other physical activity (63%), recap of key messages within the session (25%) and a cool down (0%).

Although Open Goals aims to involve the whole family in its delivery, during the observed sessions 92% (*n* = 66) of children actively participated in more than 50% of the activities, whereas only 37% (*n* = 16) of adults present took part. Although child participation rates were often very similar, adult participation numbers varied significantly across parks. Although Park A saw seven adults participating across the eight sessions, Park C had nine adult participants and Park B had no adults partaking in Open Goals activities. Coaches perceived that the level of family engagement, despite encouragement, in some parks, has affected the delivery as intended; “in some areas like [*park*] on a Friday, the parents just don’t want to get involved”.

### 3.4. Effectiveness: How Has Open Goals Influenced Park Usage and Physical Activity in the Local Area?

A total of 66 target areas from all three parks were observed, totalling 10 observations (five with OG programme offered, five without). Due to weather and time constraints, a second round of analysis could not be completed at park A.

Frequency demographics for both overall park and specific OG target area park usage and physical activity levels are reported by a cumulative total breakdown (Table 4).

Independent sample *t*-tests were used to compare the mean numbers of park users for (1) overall park usage and (2) specific OG target area usage both when OG was offered and when it was not offered. The means for overall park usage when OG was offered (M = 50.33/SD = 15.31) and when OG was not offered (M = 24/SD = 6.25) were not significantly different; t(4) = −2.76, *p* = 0.051. On the other hand, within the specific target area where OG was delivered, a significantly greater number of park users were present when OG was offered (M = 13.7/SD = 2.52) than when the programme was not offered (M = 0.33/SE = 0.57) in the same target areas; t(4) = −8.94, *p* = 0.001.

Chi-squared tests found that differences in overall park physical activity levels while the programme was offered and while it was not offered were significant (χ^2^= 16.928, *p* = 0.02, df = 2). Forty percent (*n* = 96) of those using the park when Open Goals was being offered were engaging in vigorous physical activity, compared to just 19% (*n* = 24) when Open Goals was not present.

### 3.5. Acceptability and Context: How Do Families and Coaches Perceive Open Goals?

#### 3.5.1. Bring Communities Together

All of the coaches discussed aspects where they perceived OG to bring the local community together. One coach said “it’s just to engage in communities where there is no, you know, they might not be going out playing sports and even using these parks”. Parent and child participants also recognised the importance of OG in positively bringing the community together, e.g., “it’s great, I’ve told [*other parent*] and now she’s brought her kids down this week to see what it’s all about”. Similarly, a child participant stated they “like playing with other kids from school who I wouldn’t usually play with”.

#### 3.5.2. Improve Physical, Social and Mental Health and Wellbeing

All coaches that were interviewed recognised the importance of OG in improving physical activity levels, remarking that “you’re not just working with the children, you’re working with the adults to make them more active, which will help them be more active with their family as well”. Although coaches recognised the benefit of OG in improving PA levels, they spoke more in-depth about the development of social skills and confidence within participants.

“There’s a lot of people with autism in the group, and again, they were really shy. They would never interact with other children, and now what we’re getting is, they’re sort of having little conversations and interacting with each other, and not just one-word answers, so they can now have conversations when they feel comfortable… It’s all about making those children [with autism] comfortable.”(Coach 3)

#### 3.5.3. Increased Park Usage

Family participants recognised how OG may have changed their use of the local park, stating, “I suppose it’s made us come to the park a bit more…we probably don’t make the most of the park especially as it’s so close to where we live”.

#### 3.5.4. Increased Opportunity to Access a Variety of Sports

Children and parent participants provided positive feedback about the value and impact of the sessions. For example, stating that Open Goals is, “multi-sport away from the focus of football”. The adaptability of skills learnt through OG was also a selling point; “it’s good that they’re counting as well…it’s more like skills than just sport”.

Coaches recognise the importance of offering multi-sport opportunities as part of the programme, as well as its wider aims to move away from football to a more all-encompassing sporting opportunity.

“Where it’s positioned, sport-wise, what it’s trying to do, I think...it just fits in nicely where it is, because you don’t have to be good at sport to come down to the park and get involved, because although it comes under the banner of multi-sport, we do throwing and catching, or rolling a ball, or we do quizzes and things like that”(Coach 2)

#### 3.5.5. Increase Family Time Together

It was the coaching staff that most commonly verbalised the importance of Open Goals in family co-participation;

“Open Goals…it’s unique, I think it’s got that social element of it, it’s got that family engagement element of it”(Coach 1)

“From families being able to spend more time with each other that they might not necessary spend time with them. It is brilliant to see, and I think children benefit then in so many ways”(Coach 3)

#### 3.5.6. Increase Awareness of LFC Foundation Brand

The coaches recognised that the Open Goals programme would help to raise awareness of the LFC Foundation brand, through the use of pop-up marketing and LFC Foundation branding at the parks during delivery. One coach verbalised that he thought the badge was a draw to participating in the programme, stating, “I’m a big believer in the club badge…that’s a draw isn’t it? That’s what brings the numbers in, the badge”, with another testifying, “they’re getting involved with the club [LFC] more than they were before they got involved [in Open Goals]… they’re certainly more involved with the club”.

## 4. Discussion

This study conducted a formative evaluation of the LFC Foundation’s Open Goals programme—a community-based multi-sport family PA programme. The evaluation focused on delivery of the intervention to gain an understanding of what activities were being delivered, how they were being delivered, why and to whom they were being delivered. Open Goals reached 107 participants at the three evaluation parks during the evaluation period (May–July 2019). Overall fidelity of sessions was good (69%); however, we found inter-park variation in relation to family co-participation in sessions. Dose received was measured using SOPARC and revealed positive effects on park usage and PA levels. There was a perceived high level of acceptability of Open Goals by participants and coaches. Coaches referred to families getting involved together and improvement of children’s social skills.

### 4.1. Reach

Open Goals reached 107 participants through sessions, with 423 session attendances. In total, 30.8% of participants were female and 85% were white British people. Previous studies have shown that girls are less likely to engage in PA programmes [41] and that children from some minority ethnicities engage in less vigorous PA than their white British counterparts [42]. It is important to focus on how we can increase engagement of these groups to reduce health inequalities that may result from a persistent lack of physical activity for these individuals. Further work is required to explore the barriers to participation in such groups and investigate how these could be overcome.

Open Goals sought to increase its reach by delivering sample sessions and school assemblies in schools within proximity to the park. However, although six days of sample sessions were delivered for park B, no sample sessions were delivered for parks A and C due to staffing constraints and issues with access to nearby schools. Participant numbers in park B were considerably higher than in parks A and C, indicating that sample sessions may have helped increase recruitment to the programme. The difference in whether local schools received or did not receive sample sessions and assemblies was mainly dependent on staff capacity to deliver such sample sessions. This suggests that not all areas were given equal opportunity to access their local OG sessions. Increased capacity to deliver a comparable marketing model of school assemblies and sample sessions to introduce OG before launching each park should be considered to ensure equality in recruitment [43,44].

Harkins et al. (2010) suggest that active, targeted recruitment may be more useful in socio-economically disadvantaged groups, as well as the utilisation of multiple advertising channels, such as social media campaigns and advertising in primary care and community settings, to promote the programme [43,44]. Word of mouth, as mentioned by participants, remains a highly effective means of recruitment and should be considered when marketing Open Goals. Remaining active in this recruitment process through building relationships with target groups has proven to be effective in other community-based approaches [45]. Previous recommendations have been made that suggest that capitalising on the brand (i.e., logos and imagery) and people (i.e., senior players, mascots) in order to drive a marketing campaign could help to improve participation and engagement [46]—this is another potential consideration when planning marketing strategies for the Open Goals programme.

### 4.2. Fidelity

Some discrepancies in fidelity were apparent through observation of session delivery. Although, average fidelity was 69% and therefore rated as high, there was significant variation in the delivery of the intervention as intended (see Table 3). It is important that all aspects of the OG programme are delivered as intended, in order to provide the best opportunity for participants to benefit from the programme and for OG to be effective in achieving its outcomes [47].

There was evidence of deviation in the co-participation element of the programme, with adults not participating in some sessions at some parks. Open Goals aims to include family members as part of its delivery and session observations indicated that overall, family co-participation rates were low. The level of involvement of families within the programme varied by park, potentially highlighting some important contextual influences around programme fidelity that we have not considered in this study. There is a scarcity of research suggesting why children-only may participate in so-called family-based PA interventions. The literature highlights that co-activity programmes that have been previously tried and tested have been largely unsuccessful [48]. Interestingly, Rhodes and Lim (2018) found that parents’ perceived activity differences, low interest in participating in activities or a general unwillingness to be active with the child and engage in co-activity were cited as reasons for lack of co-participation, alongside parental health-related barriers, and time-related aspects influencing parent/child co-participation [48]. It is important to recognise the potential barriers to maintaining implementation fidelity in real life contexts [49]. Involving families in the programme proved the most challenging intervention component for coaches. Future research to determine how coaches can better engage with families would be valuable for both coaches and programme developers. Coaches should recognise barriers to participation that parents may face, or perceive that they face, in order to adapt sessions to facilitate family co-participation. Previous studies suggest that combining goal setting and reinforcement techniques, for example, rewarding achievement (e.g., receipt of equipment as prizes) or recording progress, may improve levels of motivation in family PA [1]. Recognising the impact of coach education on programme delivery is also important. Several studies have explored the efficacy of brief intervention training for coaches in various aspects of health promotion [50,51]. Previous studies of community-based PA interventions highlight that coach training should go beyond ‘the typical’ football and/or multi-sport qualifications in order to accommodate the increasing needs of programme participants [46]. Improving coach communication with parents on the aims, goals and delivery of OG is recommended. Without explaining the aims to families and offering the opportunity to engage with the full intervention (improving fidelity), we cannot fully understand whether engagement or opportunity is the barrier to participation [52].

### 4.3. Effectiveness

Although the Open Goals programme itself may not have led to a significant increase in park use, the programme did generate a significant increase in park target area usage where the programme was being implemented, suggesting that potentially programme users were already park users. This may have occurred if Open Goals participants were made aware of the programme while visiting the park, through conversation with coaches or through seeing advertisement of the sessions.

Moreover, trends indicated that the presence of Open Goals in the park meant that park users were engaging in more vigorous activity during programme hours (compared to the number of people engaging in vigorous activity in the park when no programme activities were going on). Such findings are supported by other studies which recognise that community-based PA programmes can increase PA levels in parks [39].

### 4.4. Acceptability

Coaches who were responsible for delivering OG believed that the programme met their expectations and provided participants with a valuable experience. They perceived that it had significant benefits for improving children’s social skills and interactions, while bringing more families together to participate in physical activity. Although physical activity is an important outcome for OG, this is not explicitly indicated to families, with the more commonly perceived appeal being that OG should allow families to spend more time together while learning new, fundamental movement skills in local community spaces.

A recent systematic review found that focusing programmes on something other than PA for health and weight loss may be a valuable approach in terms of their success [1]. The UK Chief Medical Officer suggests that learning new skills, improving confidence, spending time together and increasing social interaction, using physical activity merely as the vehicle for such change, may be more attractive to families who do not currently meet recommended PA guidelines [3]. Open Goals should continue to be promoted as a community-based programme, with emphasis placed on other central aspects of the programme, away from merely increasing physical activity.

### 4.5. Improvements

The importance of PA for health and the development of basic movement skills warrants continued efforts to try to learn from experiences of programmes such as Open Goals, to help address difficulties in delivery and to identify ways in which PA can be incorporated more definitely within family weekend activities.

Regarding future interventions and the development of Open Goals going forward, lessons were learnt from the implementation and delivery of OG across the programme’s parks and open spaces. Recommendations from the Open Goals evaluation would be to: (i) standardise the marketing strategy of OG, particularly focusing on the model of sample sessions delivered in local schools, (ii) support continuing professional development (CPD) to include development of skills to involve families within activities together and support coaches to feel confident in speaking to families, (iii) allow some flexibility in delivery dependent on the local community and participants.

The offer of professional development to coaching staff, through Create Development™, is important in the overall delivery of OG. Research suggests that professional coach development can increase knowledge, particularly around aspects of positive youth development, focusing on positive experiences of youths as a way to build values and sustain long-term prevention of unhealthy and risky behaviours [53]. Development of excellent coaching not only makes sessions more enjoyable and structured for participants, but also motivates participants to return to subsequent sessions [54]. It is important that programme developers and coaching staff consider their approach to coaching multi-sport programmes in order to retain participants, while ensuring the outcomes of the intervention are met through the achievement of an acceptable level of fidelity.

### 4.6. Strengths and Limitations

One of the main strengths of this study is the mixed-methods nature of evaluation. Use of both qualitative and quantitative methods allowed for a thorough examination of the programme [29]. Quantitative data regarding the reach of the programme and fidelity of delivery highlight the need to modify and improve some aspects of the marketing and delivery of the intervention. Moreover, qualitative findings provide a richer description of the attitudes and opinions of coaches and anecdotal feedback from families. Coach feedback of the perceived gain of participants, particularly children, are crucial to contextualising barriers and facilitators to engagement in OG.

There are several limitations within this study. Demographic data collected by the LFC Foundation about participants was limited and had poor accuracy, particularly related to participant ages, disabilities and the number of families engaged. As this was a small-scale formative evaluation, conducted in three parks across Liverpool, the number of intervention settings was restricted and the sample size was low, limiting the generalisability of findings. Qualitative data was obtained from a small sample of participants due to the challenges in recruitment typically found in real world evaluation.

Recommendations for future research and greater infusion of qualitative information provided by participants and key stakeholders will be used to determine the wider impact of the programme. In addition to this, pre/post surveys conducted with new participants will help us to demonstrate the longer-term impact of the programme and its influence on family physical activity. It is also important to highlight to the funders the importance of collecting accurate demographic information to inform and support the research findings.

## 5. Conclusions

This research aimed to explore the implementation of Open Goals as a programme, allowing recommendations to be made for future improvements and sustainability. More specifically, it considers how effective its development and approach to delivery were at engaging families from hard-to-reach populations (i.e., those from socio-economically deprived areas and those who were physically inactive). Our findings support and enhance the idea that professional football clubs are well positioned to connect with local communities and have an important role in reaching, attracting and engaging participants in health promotion activities.

The Open Goals programme in local parks offers a promising strategy to engage families in physical activity opportunities. As research in this area is limited, more evaluations of these types of park recreation opportunities are necessary in order to strengthen our findings. Given that many football club charities have a social responsibility to support community-based programmes within their local area, those with programmes pertaining to increased park usage and physical activity should be encouraged to use such evaluation methods to evaluate interventions in their local communities.

## Figures and Tables

**Table 1 children-07-00119-t001:** Open Goals evaluation measures and data collection methods.

Measure	Definition	Method of Data Collection	Data Collected
Reach	The proportion of intended target audience that participates in an intervention ^a^	Registration forms	Number of Open Goals (OG) participants
Dose delivered	The number or amount of intended units of each intervention or each component delivered or provided ^a^	Session observation forms	Delivery of programme components for each OG session
			Information from OG coordinators on school sample session delivery and marketing strategies
Fidelity	The extent to which an intervention was delivered as planned ^a^	Session observation forms	Adherence to programme components for each OG session
Effectiveness	The extent to which participants engage with, interact with and are receptive to, and/or use materials or recommended resources ^a^	System for Observing Play and Recreation in Communities	Number of participants using parks/open spaces and their level of moderate-to-vigorous physical activity (MVPA).
Programme acceptability (participant)	To extent to which participants perceive an intervention to be suitable to their needs	Participant feedback	Satisfaction with OG programme
			Participation benefits
			Recommendations
Programme context (coach)	Aspects of the wider social environment that may influence intervention implementation	Qualitative semi-structured interviews	Perceived benefits
			Barriers to delivery
			Recommendations

^a^ Adapted from Linnan and Steckler (2002) [34].

**Table 2 children-07-00119-t002:** Description of observation record form for assessing fidelity of delivery.

Programme Component	Essential Elements	No. of Items	Format
Logistics	Session start/finish time	5	Numerical
	Number of children (+actively involved)		
	Number of parents (+actively involved)		
	Adverse events/issues		Yes/No
Welcome	Staff arrive on time	4	Yes/No
	Greeting arriving participants		
	Registration forms complete		
	Risk assessments complete		
Physical Activity	Overview and introduction to session	8	Yes/No
	Coaches outline session outcomes to participants at the start of session		
	Warm-up at beginning of session		
	Activity introduced		
	All participants involved in main activity		
	Encourage families to seek other physical activity (PA)		
	Coaches reflect on session outcomes with		
	participants at end of session		
	Cool down		
Behavioural Skills/Group Time	Parents involved in one aspect of PA with children	3	Yes/No
	Teachable moment explained/delivered		
	Demonstration of teachable moment		
Quality Checklist	The coach encourages participation	3	5-point Likert scale
	The coach reminds participants of the benefits of PA		
	The coach is enthusiastic		

**Table 3 children-07-00119-t003:** Total Open Goals park participant numbers (May–July 2019).

Park	Total Participants	Session Attendances
**Evaluation Parks**		
Park A	27	93
Park B	53	237
Park C	27	93

**Table 4 children-07-00119-t004:** System for Observing Play and Recreation in Communities (SOPARC) frequency demographics for park usage.

		Whole Park		Open Goals Target Area	
		Without Open Goals	With Open Goals	Without Open Goals	With Open Goals
**All Evaluation Parks**					
Park Usage	People	127	242	2 *	69 *
Physical Activity	Sedentary	37 (29%) *	59 (24%) *	0	14 (20%)
	Moderate	66 (52%) *	87 (36%) *	1 (50%)	0
	Vigorous	24 (19%)*	96 (40%)*	1 (50%)	55 (80%)
	MVPA _1_	90 (71%)	183 (76%)	2 (100%)	55 (80%)
Sex	Female	62 (49%)	85 (35%)	0	20 (29%)
	Male	65 (51%)	157 (65%)	2 (100%)	49 (71%)
Perceived Age	Child	28 (22%)	87 (36%)	1 (50%)	41 (59%)
	Teen	37 (29%)	39 (16%)	0	0
	Adult	58 (46%)	108 (45%)	1 (50%)	28 (41%)
	Senior	4 (3%)	8 (3%)	0	0
Ethnicity	White	120 (98%)	237 (98%)	2 (100%)	69 (100%)
	Other	7 (2%)	5 (2%)	0	0

* Significant mean differences were found at *p* <0.05. _1_ Moderate-to-vigorous physical activity.

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
