# Peer review of "Formative Evaluation of Open Goals: A UK Community-Based Multi-Sport Family Programme"

_children, 2020, doi:10.3390/children7090119_

Round 1

Reviewer 1 Report

This is a very important paper that has the potential to advance the science. I was very interesting to hear more about the community COHESION aspect of the study, but it was buried under the implementation and program evaluation. At the end, the paper reads like a prog. evaluation. I wonder if the authors could bring community cohesion upfront in the paper, or rework the title of the paper.

Also, I was not sure about Table 4, was general park use? or is it data associated with data collection. To me, it was not very clear the characteristics of those who were assessed or interviewed as part of data collection.

Author Response

Point 1

This is a very important paper that has the potential to advance the science. I was very interesting to hear more about the community COHESION aspect of the study, but it was buried under the implementation and program evaluation. At the end, the paper reads like a prog. evaluation. I wonder if the authors could bring community cohesion upfront in the paper, or rework the title of the paper.

Response

Thank you for this comment. In this phase of the evaluation there was limited evidence of the community cohesion aspects of the programme and so it is difficult to be able to include this. Subsequent qualitative enquiry has been conducted as the second phase of the evaluation which demonstrates this in more depth and we hope to publish this as a second phase of work in the future. As a response we have reworked the titled of the paper to exclude this at this point.

Point 2

I was not sure about Table 4, was general park use? or is it data associated with data collection. To me, it was not very clear the characteristics of those who were assessed or interviewed as part of data collection.

Response

To make this clearer I have omitted the word 'overall' from the table heading. To clarify, SOPARC observation looks at general park use and data was collected both when the intervention was taking place and when it was not (at similar times of the day/day of the week) to make comparison between how many people were using the park generally and the level of PA that was being undertaken. I hope that this is clear in the description of the methodology (lines 166-183).

Reviewer 2 Report

Line 67 - "Despite this, these are often the most underused parks, due to poor quality and perceptions of safety, 67 important predictors of engaging in park-based PA" This line implies that this is the predominant barrier to families using parks and whilst it is no doubt a barrier there are many others.

Line 174 - During each scan the observer recorded each  individual in view within their target area according to; their broad age group, sex and the activity they were engaged in. - The basis on which the activity classifications were made was not made clear. How was the observer able to differentiate between activity intensity?

Line 220 - Coaches also perceived that the "bad reputation" in some of the parks had an influence on participant numbers due to perceived sense of safety when using the park. - this is an assumption on which the programme was developed. What did families say?

No mention is made about sustainability of outcomes. Did the families continue to be active beyond the programme.

Word of mouth is an important marketing strategy and this is mentioned by a participant but this is not mentioned in the discussion.

Author Response

Point 1

Line 67 - "Despite this, these are often the most underused parks, due to poor quality and perceptions of safety, 67 important predictors of engaging in park-based PA" This line implies that this is the predominant barrier to families using parks and whilst it is no doubt a barrier there are many others.

Response

Thank you for this comment. I have re-worded this line to reflect that this is just one of several barriers which may influence park usage and park-based PA in families.

Point 2

Line 174 - During each scan the observer recorded each  individual in view within their target area according to; their broad age group, sex and the activity they were engaged in. - The basis on which the activity classifications were made was not made clear. How was the observer able to differentiate between activity intensity?

Response

Activity classifications were made based on the classifications used in the original SOPARC protocol. This has been revised within the article and the line 178 'In line with original SOPARC protocol [37], physical activity levels were coded as sedentary (i.e. lying down, sitting, or standing), walking, or vigorous (i.e. any activity level more strenuous than walking)', has been added.

Point 3

Line 220 - Coaches also perceived that the "bad reputation" in some of the parks had an influence on participant numbers due to perceived sense of safety when using the park. - this is an assumption on which the programme was developed. What did families say?

Response

At this point in data collection, families gave no comment on the perceived sense of safety when using parks. Following analysis of the coach interviews and the ongoing data collection through the second phase of the study (through focus groups) we questioned families on this perceived safety. We hope to publish these findings as a second impact evaluation paper in the future.

Point 4

No mention is made about sustainability of outcomes. Did the families continue to be active beyond the programme.

Response

At the point of writing, we had no evidence from families regarding their activity levels beyond the programme. As we were interested in the sustainability of the programme, we asked families about this in the second phase of the evaluation and found that the programme helped families to be more active beyond the programme. Again, we hope that these findings are published as an impact evaluation paper in the coming months.

Point 5

Word of mouth is an important marketing strategy and this is mentioned by a participant but this is not mentioned in the discussion.

Response

Thank you for this comment. Please see addition of text and relevant reference in the discussion section (lines 354-357 to reflect this); 'Word of mouth, as mentioned by participants, remains to be a highly effective means of recruitment and should be considered when marketing Open Goals. Remaining active in this recruitment process, through building relationships with target groups, has proven  to be an effective in other community-based approaches.